

# Improved chloride quantification in quadrupole aerosol chemical speciation monitors (Q-ACSMs)

Anna K. Tobler[1], Alicja Skiba[2], Dongyu S. Wang[1], Philip Croteau[3], Katarzyna Styszko[4], Jarosław Nęcki[2],
Urs Baltensperger[1], Jay G. Slowik[1] and André S. H. Prévôt[1]

[1]Laboratory of Atmospheric Chemistry, Paul Scherrer Institute, 5232 Villigen PSI, Switzerland
[2] AGH University of Science and Technology, Faculty of Physics and Applied Computer Science, Department of Applied Nuclear Physics, 30-059 Krakow, Poland
[3]Aerodyne Research, Inc, Billerica, Massachusetts, USA
[4]AGH University of Science and Technology, Faculty of Energy and Fuels, Department of Coal Chemistry and Environmental Sciences, 30-059 Krakow, Poland

*Correspondence to:* Jay G. Slowik (jay.slowik@psi.ch) and André S.H. Prévôt (andre.prevot@psi.ch)

**Abstract.** Particulate chloride is an important component of fine particulate matter in marine air masses. Recent field studies also report elevated concentrations of gas-phase reactive chlorine species and particulate chloride related to anthropogenic activities. This work focuses on particulate chloride detection and quantification issues observed for some quadrupole aerosol chemical speciation monitors (Q-ACSM), which are designed for long-term measurement of ambient aerosol composition. The ACSM reports particle concentrations based on the difference between measurements of ambient air (sample mode) and particle-free ambient air (filter mode). For our long-term campaign in Krakow, Poland, the Q-ACSM reports apparent negative total chloride concentration for most of the campaign when analyzed with the default fragmentation table. This is the result of the difference signal from $m/z$ 35 ($^{35}Cl^+$) being negative which dominates over the positive difference signal from $m/z$ 36 ($H^{35}Cl^+$). Highly time-resolved experiments with $NH_4Cl$, $NaCl$ and $KCl$ particles show that the signal response of $m/z$ 35 is non-ideal, where the signal builds up and decreases slowly for all three salts, leading to a negative difference measurement. In contrast, the $m/z$ 36 signal exhibits a near step-change response for $NH_4Cl$ during sampling and filter period, resulting in a positive difference signal. The response of $m/z$ 36 for $NaCl$ and $KCl$ is not as prompt as for $NH_4Cl$ but still fast enough to have a positive difference signal. Furthermore, it is shown that this behavior is mostly temperature-independent. Based on these observations, this work presents an approach to correct the chloride concentration time series by adapting the standard fragmentation table coupled with a calibration of $NH_4Cl$ to obtain a relative ionization efficiency (RIE) based on the signal at $m/z$ 36 ($H^{35}Cl^+$). This correction can be applied for measurements in environments where chloride is dominated by $NH_4Cl$. Caution should be exercised when other chloride salts dominate the ambient aerosol.



# 1    Introduction

Aerosols are known to have a significant influence on regional to global climate and visibility (Fuzzi et al., 2015). Furthermore, severe adverse health effects have been linked to aerosol exposure (Pope and Dockery, 2006). Therefore, a better understanding of the aerosol composition is crucial. In recent years, different types of Aerodyne aerosol mass spectrometers (AMS (Jayne et al., 2000)) and aerosol chemical speciation monitors (ACSM (Ng et al., 2011)) have been widely used to quantify the chemical composition of non-refractory (NR) particulate matter (PM) with high time resolution. They allow simultaneous quantification of NR-PM chloride, ammonium, nitrate, organics and sulfate. For many environments, chloride does not significantly contribute to the total mass (Jimenez et al., 2009). Exceptions include coastal regions influenced by marine aerosol masses which are rich in inorganic sea salt (Ovadnevaite et al., 2012) and polluted inland regions influenced by anthropogenic chloride emissions. Anthropogenic chloride emissions include HCl which then forms $NH_4Cl$ with ammonia ($NH_3$). Particulate chloride can enable heterogeneous production of $ClNO_2$ via reactive uptake of $N_2O_5$ during nighttime, which photolyzes to produce highly reactive chlorine radicals in the daytime (Yang et al., 2018; Le Breton et al., 2018). The chlorine radical undergoes hydrogen-abstraction reactions with hydrocarbons to form HCl which then again forms $NH_4Cl$ (Chang and Allen, 2006). This can lead to exceptionally high chloride concentrations as has been reported for New Delhi by Gani et al. (2019) and Tobler et al. (2020).

ACSMs equipped with a quadrupole mass spectrometer (Q-ACSM, (Ng et al., 2011)) have been used in numerous field campaigns in various environments. The first ambient intercomparison of 13 individual Q-ACSMs, carried out in the vicinity of Paris, revealed good correlation between the Q-ACSMs for all species except chloride. It was not clear if this was due to low concentrations near the detection limit or due to the non-ideal vaporization behavior of chloride, which depends on the specific conditions in the individual instruments (Crenn et al., 2015). During our long-term (> 1 year) measurement campaign in Krakow, Poland, we observed significant apparent negative chloride concentrations, especially during the winter season when aerosol concentrations are high in general. The fragmentation table (Allan et al., 2004) attributes each $m/z$ to one or more of the bulk species, i.e. nitrate, ammonium, sulfate, organics and chloride. By default, the chloride concentration is calculated based on the measured $^{35}Cl^+$ ($m/z$ 35) and $H^{35}Cl^+$ ($m/z$ 36) ion signals and the estimated $^{37}Cl^+$ ($m/z$ 37) and $H^{37}Cl^+$ ($m/z$ 38) ion signals, calculated based on the natural isotope ratio of chlorine (see Table 1). Measurements with negative signal from either $m/z$ 35 or 36 can therefore result in total negative chloride concentrations. The behavior of some chloride salts in the AMS has been studied. For example Ovadnevaite et al. (2012) demonstrated that sea salt can be quantified by an AMS despite the mostly refractory nature of NaCl. Drewnick et al. (2015) showed that chloride salts appear to be sticky on the vaporizer surface and are only slowly removed. Also, chloride can undergo chemical reactions with the tungsten vaporizer surface resulting in $WO_2Cl_2$, among other compounds. Furthermore, they showed that chloride detection suffers from vaporizer memory effects, as demonstrated by the presence of several iron chloride signals when iron nitrate nonahydrate was injected after chloride experiments. For the Q-ACSM, detection and quantification issues for organic chloride have been reported for isoprene-



derived secondary organic aerosol (Wang and Hildebrandt Ruiz, 2017). To our knowledge, there are no studies focused on the response of Q-ACSM to inorganic chlorides.

In this study, we are able to attribute the issue of reported negative chloride concentrations in Q-ACSMs to a negative difference signal at *m/z* 35. We present a highly time-resolved characterization of the Q-ACSM response to three different chloride salts at different vaporizer temperatures. Finally, we propose a calibration procedure coupled with a change in the fragmentation table to improve chloride detection and quantification using the Q-ACSM.

## 2    Method

The operating principle of the ACSM is described in detail by Ng et al. (2011) and is briefly summarized here. The ACSM alternatively samples directly from ambient air ("sample") and through a particle filter ("filter"), switching every 30 s. After passing through a 100 μm critical orifice, the submicron particles are focused into a narrow beam by an aerodynamic lens. Non-refractory particles are flash-vaporized upon impact with the standard tungsten vaporizer at ~600 °C, where it is assumed that the solid particle is quickly turned into a vapor without undergoing any other processes besides thermal decomposition.

For the ACSM used in this study (SN 140-145), an yttriated iridium filament was used to ionize the resulting vapors via electron impact (EI). The ions are detected by a quadrupole residual gas analyzer (RGA, Pfeiffer Vacuum Prisma Plus). The difference of the sample and the filter measurements represents the aerosol mass spectrum. The obtained mass spectrum typically ranges between *m/z* 10 and 150 with unit mass resolution (UMR).

Investigations with different high-purity chloride salts were performed. Ammonium chloride ($NH_4Cl$), sodium chloride (NaCl)
and potassium chloride (KCl) were separately dissolved in ultrapure water. Mono-disperse particles with a diameter of 300 nm were generated using a Topas atomizer, a krypton source as a bipolar charger and a custom-made differential mobility analyzer (DMA). The particles were simultaneously injected into the ACSM and a condensation particle counter (CPC, TSI model 3022A). The standard vaporizer voltage was set to 7.7 V, corresponding to a vaporizer temperature of 720 °C, throughout the full campaign and was not changed unless otherwise mentioned. Calibrations were performed with the same
setup using aqueous solutions of ammonium nitrate ($NH_4NO_3$), ammonium sulfate (($NH_4$)$_2SO_4$) and ammonium chloride ($NH_4Cl$), following the new recommended calibration procedure, which measures in full scan mode, meaning that the same scanning protocol as during ambient measurements is used during the calibration (Freney et al., 2019).

The presented ambient online measurements were conducted at the AGH University of Science and Technology in Krakow, Poland (50°04' N, 19°55' E) between 8 January 2018 and 10 April 2019. The inlet was installed 2 m above the rooftop of the
90 building and was equipped with a 5 L min$^{-1}$ PM$_{2.5}$ cyclone (BGI, Mesa Labs, Inc.). The aerosol was dried through a Nafion dryer in the temperature-controlled room before being sampled by the ACSM. The data was recorded with a resolution of 10 min unless specified otherwise. All data were analyzed using ACSM Local 1.6.1.3 (Aerodyne Research Inc.) in Igor 6.37 (Wavemetrics Inc.).


## 3    Results and discussion

### 3.1   Observations in ambient measurements

The Q-ACSM was installed at AGH University in Krakow for > 14 months. The time series of chloride and the other NR-PM species are shown in Fig. 1a and 1b. During the first month of the measurements, significant positive chloride concentrations were measured. However, shortly after, the contribution of chloride decreased and, from the end of February 2018 until the end of the campaign, apparent negative chloride concentrations were recorded.

The chloride concentration is calculated based on the $m/z$ 35 ($^{35}Cl^+$), $m/z$ 36 ($H^{35}Cl^+$), $m/z$ 37 ($^{37}Cl^+$), and $m/z$ 38 ($H^{37}Cl^+$) signals (Table 1). The $^{35}Cl^+$ and $H^{35}Cl^+$ are measured, whereas the $^{37}Cl^+$ and $H^{37}Cl^+$ signals are calculated based on the natural isotopic ratio of $^{35}Cl$ to $^{37}Cl$ and therefore will not be discussed further. While the signal of $m/z$ 36 is positive throughout the full campaign (Fig. 1c), the signal of $m/z$ 35 turns negative in February 2018 (Fig. 1d), which results in an apparent negative concentration of the total chloride signal (Fig. 1b). Similar observations of apparent negative chloride concentrations have been seen in Măgurele, Romania, during long-term field measurements (L. Marmureanu, personal communication).

Under typical operating conditions, the Q-ACSM scans the mass range from $m/z$ 10–150 at a scan rate of 200 ms amu$^{-1}$, which produces a full mass spectrum roughly every 30 s. To better understand the transient behavior of $m/z$ 35 and $m/z$ 36, only those two ions were scanned with high time resolution, leading to a signal with 1 s resolution. To simulate the typical ambient ACSM sample/filter switching, the filter was switched every 30 s. Results shown in Fig. 2 explain how the different response times of $Cl^+$ and $HCl^+$ result in the apparent negative chloride: While $HCl^+$ behaves nearly ideally and the signal instantaneously reacts after the filter change, $Cl^+$ slowly builds up or slowly decays in the 30 s following a filter change. During normal full-spectra scans, the ions are subsequently measured in the quadrupole, meaning that the $Cl^+$ and $HCl^+$ signals used to calculate the difference signal are measured approximately 5 s after the filter switch, as indicated by the markers in Fig. 2. Consequently, the difference signal (i.e. "sample" – "filter") for $m/z$ 36 is positive, whereas the difference signal for $m/z$ 35 is negative under normal operating conditions.

### 3.2   Behavior of selected chloride salts in the Q-ACSM

The behaviors of the slowly vaporizing $Cl^+$ and the rapidly vaporizing $HCl^+$ of three of the most abundant chloride salts in the atmosphere were studied in more detail using $NH_4Cl$, $NaCl$ and $KCl$. Similarly to the highly time-resolved targeted $m/z$ measurements for the ambient sample, only $m/z$ 35 and $m/z$ 36 were monitored on a 1 s resolution basis with filter switching every 30 s.

In Fig. 3 the averaged temporal development of the signal in sample and filter mode for each of the three chloride salts is shown. The signals for $NaCl$ and $KCl$ are normalized to the signal of $NH_4Cl$. There are apparent differences between $NH_4Cl$ and the other two salts, as well as between the $m/z$'s. The signal for $NH_4Cl$ responds faster after the filter switching, particularly for $m/z$ 36 where a prompt increase and decrease of the signal after the filter switch can be observed. In contrast, the signal of $NaCl$ and $KCl$ evolves much more slowly. This different response time between the salts is also observed for $m/z$ 35, however



to a much smaller extent. The observation of the $m/z$ 36 (HCl$^+$) signal for NaCl and KCl, even though their direct thermal composition products do not include HCl(g), is probably the result of heat-induced chemical reactions between chloride and background water vapor (Drewnick et al., 2015).

In addition, the behavior of the different chloride salts at different vaporizer voltages (i.e. temperatures) was studied. The

relative temporal evolution is mostly independent of the vaporizer temperature for all three salts investigated. The temporal behavior of the signal of NH$_4$Cl is shown in Fig. 4. Similar to the usual vaporizer temperature of this instrument (7.7 V), the signal at $m/z$ 35 evolves much more slowly than the signal at $m/z$ 36 at all temperatures. Near identical temporal trends are observed for the signal at $m/z$ 36 for vaporizer voltages below 7.7 V. Above this voltage, a jump in the ion baseline intensity is observed. For the signal $m/z$ 35 a similar trend is visible, however, the differences between the signals for vaporizer voltages

$\leq$ 7.7 V are larger. For NaCl and KCl a similar trend is observed (Fig. S1 and S2), where the background signal for NaCl is already starting to build up at 7.7 V. The signal of the background is a combination of several processes that can be expected to be enhanced or suppressed by the higher temperature, including increased flash vaporization of chloride at vaporizer surface, suppressed condensation at/near the vaporizer surface, enhanced re-desorption at/near vaporizer surface, and condensation and re-desorption near the filament. To which extent each of these processes contribute to the jump in the instrument background

cannot be decoupled based on the available measurements.

In the AMS (and therefore also in the ACSM), NH$_4$Cl is expected to undergo thermal decomposition via the reaction NH$_4$Cl(s) $\rightarrow$ NH$_3$(g) + HCl(g) ((Hu et al., 2017) and references therein). This flash-vaporizing dissociation pathway results in signal at $m/z$ 36 from HCl$^+$. However, it is also possible that the particles can remain on the vaporizer or bounce off the vaporizer and land on a nearby, cooler surface, e.g., on the ionization chamber walls, and vaporize at a slower rate. During this process,

further chemical reaction may occur, e.g. the vaporizer surface can act as catalyst for reactions with other aerosol components or material on the vaporizer or the vaporizer material itself. For instance, production of different tungsten oxide chlorides are reported for a porous tungsten vaporizer (standard vaporizer) (Drewnick et al., 2015). Regardless of the reactions, any of these multi-step processes would be much more likely to suppress the $m/z$ 36 signal and enhance the $m/z$ 35 signal. For example, it can be expected that tungsten oxide chlorides, produced by reaction with the vaporizer, will result in a Cl$^+$ signal rather than

an HCl$^+$ signal.

The quantification of ACSM (and AMS) data relies on the imperfect assumption that all measured particles flash-vaporize on the initial impact with the vaporizer. However, the behavior of some compounds such as ammonium sulfate and certain organic molecules can deviate from that of an ideal non-refractory component (Huffman et al., 2005). The behavior of semi-refractory compounds can strongly depend on the instrument history, vaporizer temperature, instrument tuning, filament material, and

the physical alignment of the filament and is therefore hard to predict. The temperature in the ionization chamber is influenced by the vaporizer temperature itself, the filament temperature, and the alignment of those to each other. Nowadays, ACSMs are equipped with tungsten filaments. However, older ACSMs were delivered with yttriated iridium filaments, which is also the case for the ACSM used here for all measurements. There are no direct measurements available to compare the temperature of the ion source between these two systems. However, based on the material properties, the iridium filament is expected to



have a lower temperature compared to the tungsten filaments and therefore the iridium filament is expected to have more slow-vaporizing components compared to the tungsten filament. For example, apparent negative chloride concentrations were reported for the Romanian ACSM mentioned earlier when operated with an iridium filament. After changing to a tungsten filament, the total chloride concentration was positive (L. Marmureanu, personal communication). Based on the observations, it is also possible that the capture vaporizer (Hu et al., 2017; Xu et al., 2017) can increase the possibility of negative $m/z$ 35,

as there is more collision of HCl(g) with the hot vaporizer surface which could result in more $m/z$ 35 signal.

Regardless of how consistent the effect is between different Q-ACSMs, the fact that a difference between $m/z$ 35 and $m/z$ 36 regarding the vaporization times can be observed, suggests that the $m/z$ 35 signal should be utilized with caution even when negative difference signals are not detected. Analogous to Ovadnevaite et al. (2012), where the $NaCl^+$ ion was suggested as a surrogate for sea salt due to its more rapid evaporation, here we recommend to use the $HCl^+$ ion as the signature for $NH_4Cl$, as

it is the direct product of thermal decomposition and less influenced by secondary, lower-temperature vaporization. It also leads to more consistent results over time as the time-response of $Cl^+$ at $m/z$ 35 is hard to predict as described above.

### 3.3  Corrections

The chloride calculation is based on the frag_chloride entry in the fragmentation table (Allan et al., 2004), which is actually a combination of frag_Cl and frag_HCl (Table 1, in black). Our experimental results suggest that the $Cl^+$ signal originates from

slow thermal decomposition of $NH_4Cl$ and biases the calculation of the total chloride concentration. HCl, the thermal decomposition product of $NH_4Cl$, also fragments into $Cl^+$, however, this is calculated based on the $HCl^+$ signal. Therefore, the $Cl^+$ signal from frag_Cl should not be used in the calculation and we suggest to adapt the fragmentation table by multiplying the frag_Cl by zero (Table 1, in red). Similar suggestions were made for the quantification of organochlorides (Wang and Hildebrandt Ruiz, 2017).

As described by (Ng et al., 2011), the mass concentration $C_s$ of the species $s$ is calculated from the ion signals $I$ at its mass spectral fragments $i$, taking into account the molar weight $MW_s$ and the ionization efficiency (IE) of the species, the volumetric sample flow $Q_v$, Avogadro's number $N_A$ and a conversion factor of $10^{12}$:

$$C_s = \frac{10^{12} \cdot MW_s}{IE_s \cdot Q \cdot N_A} \sum_{all\ i} I_{s,i} \tag{1}$$

The slower detection electronics of a Q-ACSM do not allow a direct measurement of the IE, but the response factor (RF) of

the instrument is related to the IE through Avogadro's number $N_A$, the molar mass $MW$, the flow $Q_{cal}$ and the electron multiplier gain $G_{cal}$:

$$IE_{NO_3} \cdot \frac{N_A}{MW_{NO_3}} = \frac{RF_{NO_3}}{Q_{cal} \cdot G_{cal}} \tag{2}$$

Instead of determining the $IE_s$ for each species, it is more convenient to express the $IE_s$ relative to the IE of $NO_3$ ($IE_{NO3}$) as the so-called relative ionization efficiency ($RIE_s$) for each species.





$$\frac{IE_s}{MW_s} = RIE_s \cdot \frac{IE_{NO_3}}{MW_{NO_3}}$$
(3)

Routinely, the ACSM is calibrated with $NH_4NO_3$ and $(NH_4)_2SO_4$ to determine the $RF_{NO3}$, $RIE_{NH4}$ and $RIE_{SO4}$. Whereas the $RF_{NO3}$ is based on $m/z$ 30 and 46, the $RIE_{NH4}$ and $RIE_{SO4}$ are based on all ion signals of the species. In order to quantify the chloride mass properly, the ACSM is also calibrated with $NH_4Cl$ to determine the RIE_Chl'. While the standard $RIE_{Chl}$ is based on all ion signals for chloride assigned in the standard fragmentation table, we firstly adapted the fragmentation table as

described above for the calculation of the RIE_Chl', meaning that the RIE_Chl' is only based on the ion signals of frag_HCl and does not include frag_Cl. Details on the calculation of the RIEs can be found in the supplement section S1. Five calibrations over the course of 7 months (October 2018 – April 2019) resulted in an average RIE_Chl' = $0.41 \pm 0.17$, $RIE_{NH4}$ = $2.43 \pm 0.58$ and $RIE_{SO4}$ = $0.38 \pm 0.11$ with $RF_{NO3}$ = $4.68 \pm 1.66 \cdot 10^{-11}$ amps (µg m$^{-3}$)$^{-1}$. In general, the calibrated RIE values are lower than the default values commonly used in the ACSM. Notably, the RIE_Chl' is significantly lower than the default value of 1.3.

The ACSM is recommended to be routinely calibrated not only with $NH_4NO_3$ but also with $(NH_4)_2SO_4$, because it has been shown that the $RIE_{SO4}$ value can be quite different from the default value of 1.2 (Budisulistiorini et al., 2014; Crenn et al., 2015; Freney et al., 2019), as it is also the case for this instrument.

The relation of $m/z$ 36 to the total chloride mass depends strongly on the chloride salt present, as discussed in Section 3.2. $NH_4Cl$ exhibits a prompt signal response at $m/z$ 36, whereas the NaCl and KCl signals build up more slowly. It still results in

a slightly positive difference signal for those two chloride salts. However, the ratio of $m/z$ 36 to the total chloride mass will be different. Therefore, in the absence of a single dominant cation, quantification should be treated with care due to the effect of the salt-dependent vaporization kinetics on the $m/z$ 36 difference signal.

Comparison of the highly time-resolved chloride salt calibrations with ambient measurements, as well as the correlation of $m/z$ 36 with $NH_4^+$ ($R^2 = 0.58$), $m/z$ 23 (Na$^+$, $R^2 = 0.37$) and $m/z$ 39 (K$^+$, $R^2 = 0.13$, though possibly influenced by $C_3H_3^+$ ions) suggest

that $NH_4Cl$ was likely the dominant fine chloride species in the ambient aerosol in Krakow, Poland. The correlation of $m/z$ 36 and $m/z$ 58 (potentially NaCl$^+$, $R^2 = 0.71$) and $m/z$ 74 (potentially KCl$^+$, $R^2 = 0.79$) is high, though this is likely the result of correlation of $m/z$ 36 with total organics ($R^2 = 0.60$), which could produce ions at the same nominal $m/z$ (e.g. $C_3H_6O^+$, $C_2H_2O_2^+$ and $C_4H_{10}^+$ at $m/z$ 58, and $C_6H_2^+$, $C_3H_6O_2^+$ and $C_4H_{10}O^+$ at $m/z$ 74), which cannot be separated from metal halide ions with UMR data. Therefore, application of the above described correction of the fragmentation table and calibration should yield

accurate quantification of chloride. Fig. 5 shows the corrected chloride time series.

Time-of-flight instruments like the ToF-ACSM (Fröhlich et al., 2013) and ToF-AMS (Drewnick et al., 2005; DeCarlo et al., 2006) typically do not suffer from negative $m/z$ 35 signal, even though similar vaporizer and ionizer configurations are used, due to the different measurement technique of the detector (no scanning over the full mass range). However, one would expect to measure roughly 0 at $m/z$ 35 since the average value of the sample rise and the filter decay are about equal. Therefore, the

calculation methodology presented here can provide more accurate/repeatable quantification regardless of the instrument.



## 4    Conclusions

Apparent negative chloride concentrations were measured during a long-term campaign in Krakow, Poland resulting from slow vaporization of $NH_4Cl$ at $m/z$ 35, when using the standard ACSM fragmentation table. Highly time-resolved measurements of different chloride salts confirm a different behavior of $^{35}Cl^+$ ($m/z$ 35) and $H^{35}Cl^+$ ($m/z$ 36). $m/z$ 36 shows a

prompt signal response, whereas $m/z$ 35 responds more slowly which may lead to a negative difference signal for that ion. The extent to which this happens can strongly depend on instrument history, tuning and alignment in the ionizer cage and is hard to predict. Even when an instrument is not apparently affected by negative $m/z$ 35 signal, one should consider using the revised fragmentation table presented here for chloride along with an instrument specific RIE_Chl' so that the total chloride mass is calculated only based on the $m/z$ 36 signal.

Q-ACSM users should consider modifying the fragmentation table and, when doing so, include $NH_4Cl$ in routine calibrations throughout the campaign. The RIE_Chl' value of 0.41 presented here should be considered as a guidance and is only valid for this particular instrument. We suggest that routine calibration with $NH_4Cl$ be utilized to determine this value for a particular instrument when better quantification of chloride is desired. Future ACSM intercomparisons will provide an opportunity to study this issue in more detail.


*Data availability.* The data presented in the text and figures as well as in the supplement will be available upon publication of the final manuscript (https://zenodo.org). Additional related data can be made available upon request.


*Competing interests.* Philip Croteau was employed by Aerodyne Research Inc. when the experiments were conducted. The other authors declare that they have no conflict of interest.

*Author contributions.* AKT and AS carried out the data collection and the instrument calibration. AKT analyzed the data and

wrote the manuscript. ASHP, JGS, JN, KS and UB were involved with the supervision. ASHP, DSW, JGS, PC and UB assisted in the interpretation of the results. All co-authors contributed to the paper discussion and revision.

*Acknowledgement.* This work was financially supported by the EU Horizon 2020 Framework Programme via the ERA-PLANET project SMURBS (grant agreement no. 689443) and the COST action CA16109 Chemical On-Line cOmpoSition

and Source Apportionment of fine aerosoLs COLOSSAL grant. JGS acknowledges support from the Swiss National Science Foundation (starting grant BSSGI0_155846).
We sincerely thank Leah R. Williams, Manjula R. Canagaratna and John T. Jayne for the discussion about technical details of the instrument and the agreement of the presented correction within other Aerodyne mass spectrometers.





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





**Table 1. The original fragmentation table (black) is a combination of the frag_Cl and frag_HCl. Because of the non-ideal behavior**
**of *m/z* 35, it is recommended to adapt (red) the fragmentation table for chloride, so that it is only based on frag_HCl.**

| *m/z* | frag_chloride | frag_Cl | frag_HCl |
|---|---|---|---|
| 35 | frag_HCl[35],**0***frag_Cl[35] | 35,-frag_HCl[35] | 0.231*frag_HCl[36] |
| 36 | frag_HCl[36] | | 36,-frag_air[36] |
| 37 | frag_HCl[37],**0***frag_Cl[37] | 0.323*frag_Cl[35] | 0.323*frag_HCl[35] |
| 38 | frag_HCl[38] | | 0.323*frag_HCl[36] |


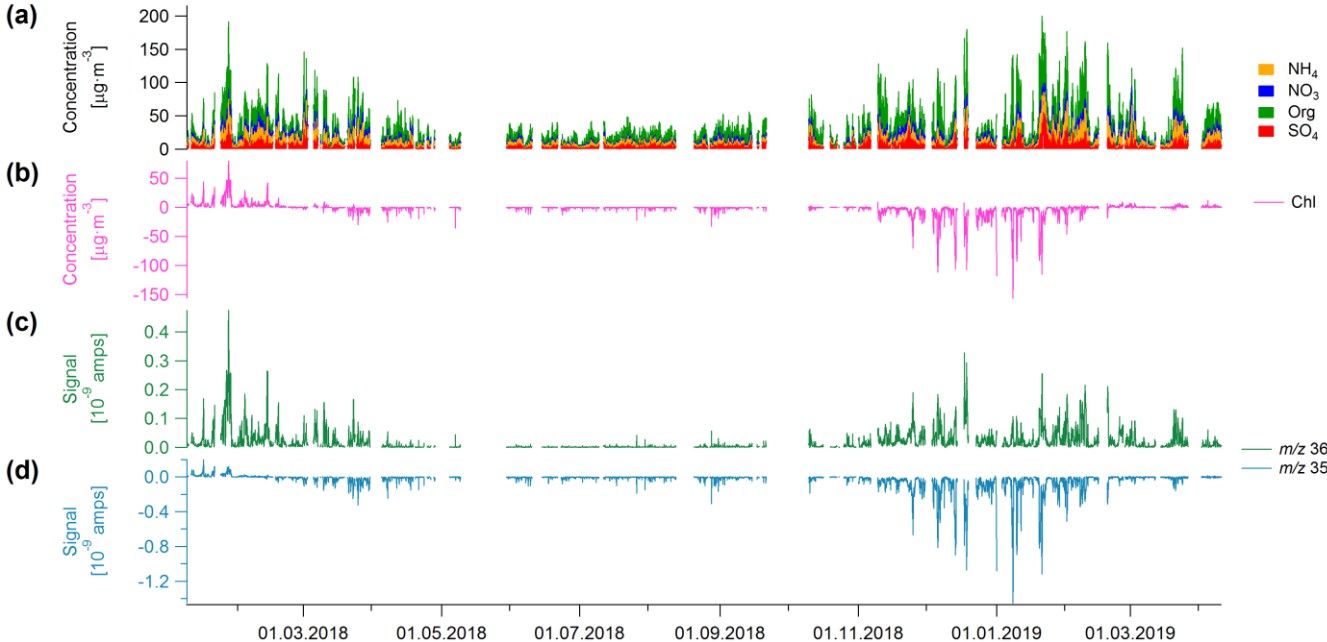

**Figure 1. (a) Stacked time series of NH₄, NO₃, Org and SO₄ and (b) time series of chloride in µg m⁻³. Panels (c) and (d) show the time series of *m/z* 35 and *m/z* 36 in amps, respectively. The negative chloride signal is driven by the negative *m/z* 35 signal.**


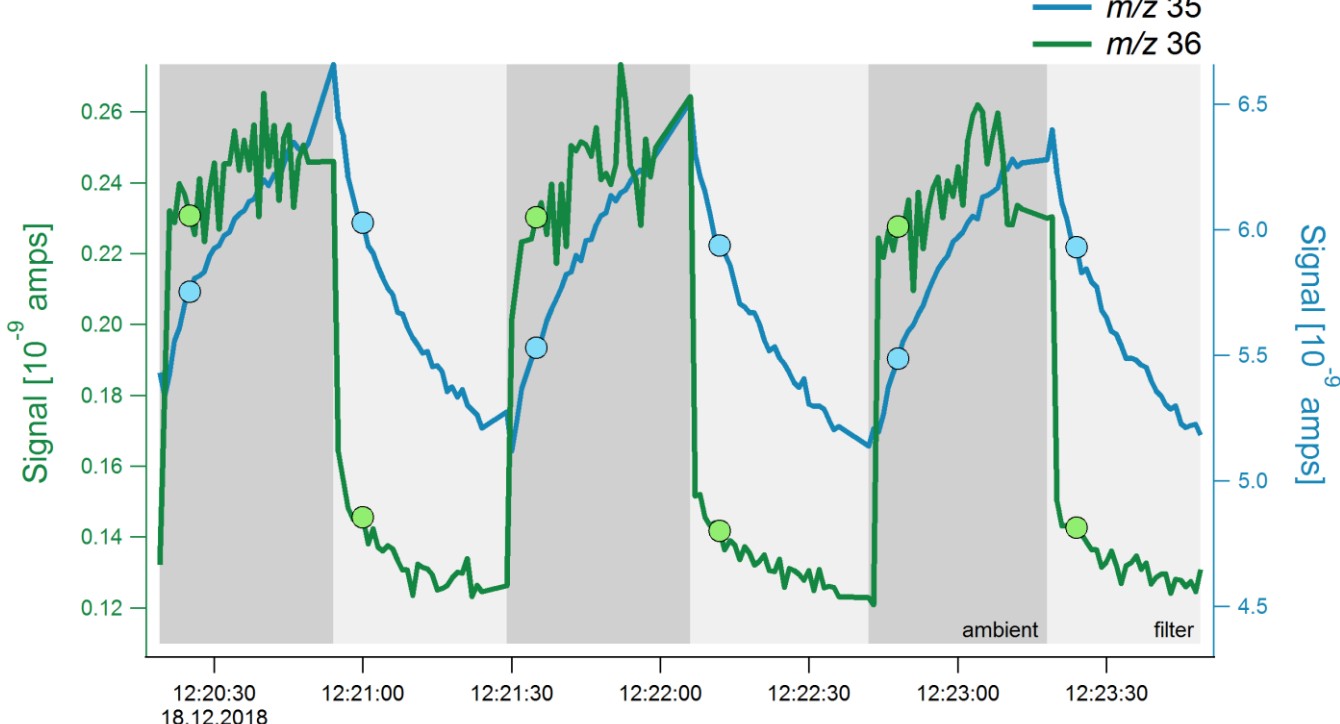

**Figure 2. High–resolution time series (1 Hz) of the total ambient signal at *m/z* 35 (blue) and *m/z* 36 (green) signal. The filter was switched every 30 s to simulate normal measurements, the filter and ambient mode are indicated by the different shades of grey. The dots mark the time at which these *m/z*'s are scanned in the quadrupole during normal measurements, which is typically around 5 s after the filter switch.**






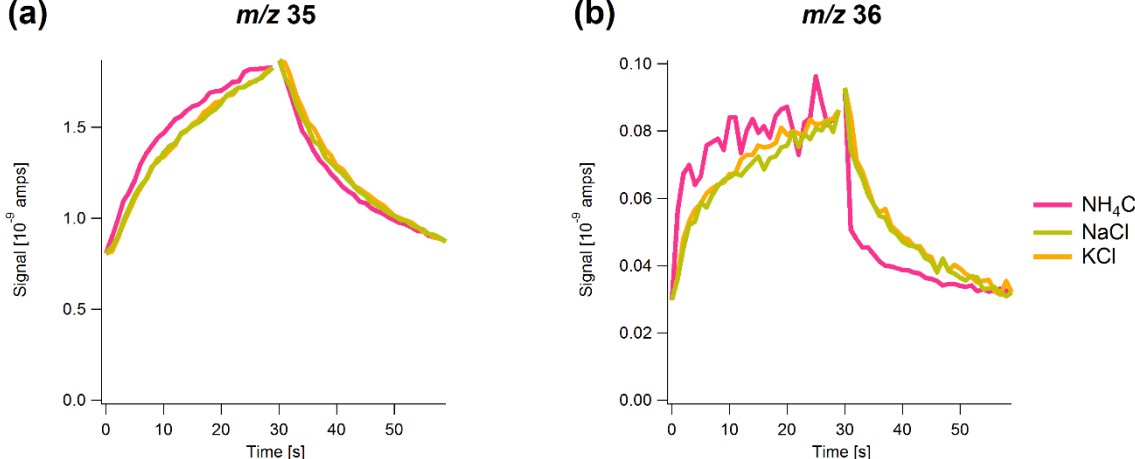

**Figure 3.** Time series of the signal of (a) *m/z* 35 and (b) *m/z* 36 with 1 s resolution over a simulated sample/filter cycle for NH₄Cl (pink), NaCl (green) and KCl (orange). The signal of NaCl and KCl are normalized to the one of NH₄Cl.






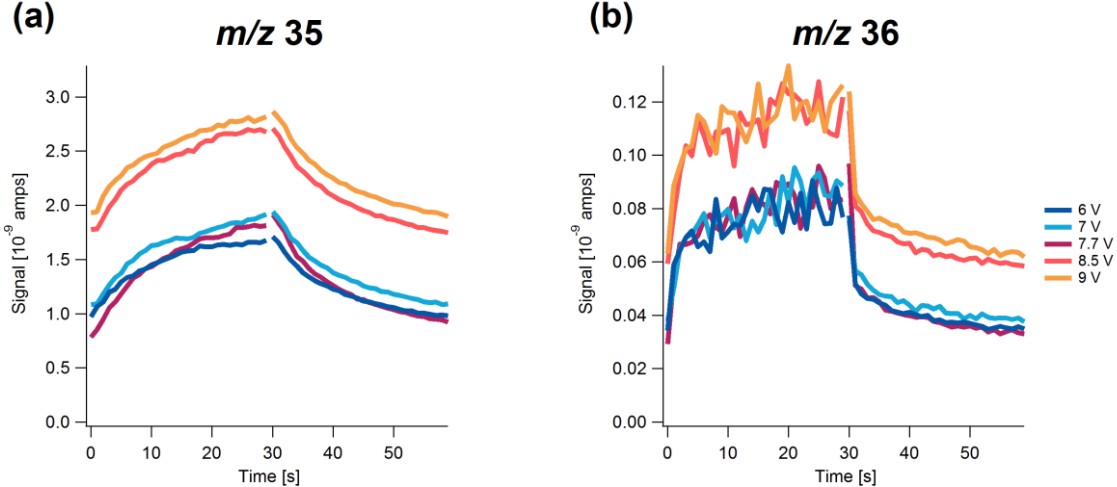

**Figure 4. Highly time-resolved signal of (a)** *m/z 35* **and (b)** *m/z 36* **as a function of time at different vaporizer voltages (i.e. temperatures) over a simulated sample/filter cycle for NH₄Cl.**



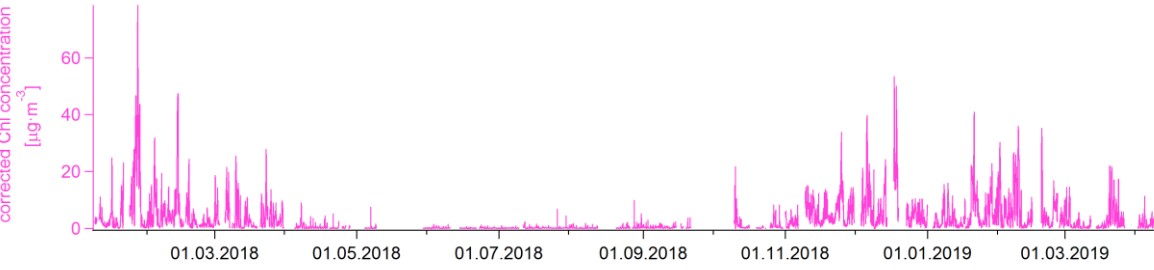


**Figure 5. Time series of chloride after recalculation based on fragmentation table adjustments and RIE_Chl' derived from NH₄Cl calibration.**