# Peer review of "Improved chloride quantification in quadrupole aerosol chemical speciation monitors (Q-ACSMs)"

_Atmospheric Measurement Techniques, 2020_

## Referee Comment (RC1) · Anonymous Referee #2 · 3 Jun 2020

General Comments:

The manuscript by Tobler et al. focused on particulate chloride detection and quantification issues observed for some quadrupole aerosol chemical speciation monitors, which presented an approach to correcting the chloride concentration. This is an important and necessary work, which can be applied for measurements in environment where chloride is dominated by NH4Cl Overall the paper is well written. I recommend acceptance for publication on AMT after minor revisions.

Specific comments:

1, What is the reason for setting the voltage to 7.7 V (line 83, 84)? Please elaborate it.

2, Please consider placing the high resolution peak fit of m/z 23, 39, 58 and 74 (line 208-215) in the supplementary.

3, Before March 2018, the chloride shows the positive concentration (Fig.2), please compare this with the chloride after recalculation based on fragmentation table adjustments, and elaborate the error margin.

4, I suggest that the sample/filter cycle in Fig.3 and 4 be shaded as an indicator, just like Fig.2.

---

## Referee Comment (RC2) · Anonymous Referee #1 · 3 Jun 2020

The manuscript provides the pathway for chloride quantification in the ACSM measurements. It is very timely and needed paper as, currently, despite the caveats listed in the paper, the Cl quantification is normally taken for granted. The manuscript is very well structured and nicely written, real pleasure to read. However, I have a major concern with the lack of method validation. The corrected Cl data were not compared or validated to anything. Indeed, the signal does become positive after the corrections, however, there is no indication that that positive signal is quantitative. Since it is very important to show that the method works quantitatively, I would strongly suggest including a corrected Cl comparison with an independent measurement. Speciated Cl measurements would be ideal, but might not be readily available, so, at least,

an improvement of total volume/mass measured with ACSM and independent instrument/instruments should be shown. Subject to this validation, I deem this manuscript suitable for the AMT.

Another important aspect, but, maybe, not as crucial as the one above, is higher than standard vaporiser temperature for this instrument. Why was this implemented, how does this compare to the standard t-re measurements? There are some indications, that 720C might still be comparable to standard 600C for some m/z, but better discussion around this is required. With some information on why this temperature was selected provided in the methods section as well. Were ambient measurements performed at this temperature as well?

Finally, the assumption that it was indeed NH4Cl contributing to Cl signal is still not fully convincing. Better discussion on NH4Cl origins and potential sources in this region is required, also discussing the potential lack of other salts (why other salts are not likely) in the region. Correlation with Na is still significant, why?

Minor: Line 79: provide details for salts (sources, purity);

Line 82: provide info for drying (type of dryer, humidity after drying, RH stability);

Line 148: 'likely to suppress the m/z 36 signal and enhance the m/z 35 signal' – do you mean background, not diff? specify.

Lines 197 very high ($\sim$40%) variability of RIEcl, was that temperature dependent? Discuss it. Also, discuss the exceptionally low RIEnh4. Was humidity stable during calibrations?

---

## Referee Comment (RC3) · Anonymous Referee #3 · 5 Jun 2020

This manuscript describes the presence of a negative chloride signal measured in the widely used aerosol chemical speciation monitor (ACSM) instrument. This is an issue that has been observed in several instruments and up to now has not been formally addressed. In this work, the authors present long term observations of this artefact and perform additional detailed tests on instrument performance. The authors illustrated than this negative signal is essentially an artefact (stating that no other information regarding the source of Cl can be extracted), and propose a simple correction to the standard fragmentation table to account for it. Given the widespread use of the ACSM, this type of work is essential to providing homogenous measurements among all operating instruments.

[Figure]

This manuscript is well written with clear and concise text and well-presented figures. Although I recommend the manuscript for publication, I have some questions and comments below.

1) It is stated in the manuscript that there are a number of uncertainties related to how this artefact manifests itself in different instruments. The authors cite a personal communication whereby changing out the filament removes this artefact. Can the authors provide more information on this; does the artefact return after some time or is this artefact only present in instruments with an iridium filament?

2) Was this instrument newly installed at the start of sampling. Could the inversion of Cl be a result of the build-up of material (the total PM concentrations observed during the field campaign are very high)? The appearance of the negative m/z-35 was very sudden, did it correspond to any changes in meteorological conditions?

3)A slow decay and slow build-up (as well as an artefact that disappears when the filament was changed) would suggest that material is built up on the vaporizer and the surrounding area. Was the filament changed (in this instrument) after the experiment to investigate this? Do you have an approximate temperature range for your experiments (that correspond to the voltages used)? Were any improvements observed after heating the vaporizer to > 800 C over extended periods of time?

Line 110 (Figure 2): Is it possible to change the instrument settings so the sampling periods correspond to the end of the filter and sample run. This would better represent species that slowly build up and slowly decay?

4)Why is the Chloride (m/z 35) signal in the negative so much larger than in the positive? As is observed in the latter part of 2018 and early 2019.

At the very end of the sample period, it appears that the total reported Cl returned positive again, is this the case?.

5) During the 14 month sampling period what other instruments were sampling alongside the ACSM, e.g. number and size distribution, filter measurements etc. Were any complementary measurements of refractory species made during this time.

6) How did the measured ACSM total mass compare with the total mass measured by the SMPS (if present) during these sample periods (excluding the negative chloride peaks periods)? Are there indicators of the presence of refractory species during this time.

7)Line 213: were any correlations observed between m/z 35 and Na+ (m/z 23) and or K+ (39)?, could these peaks also have interference with species other than NaCl+ and KCl+? When the correction is applied, is all the NH4 measured accounted for by that predicted from Cl-, NO3-, and SO42- (in the form of NH4Cl, NH4NO3, and (NH4)2SO4 respectively).

8) What recommendations should be given to data that is already submitted to data sets (e.g EBAS)?

9)Given the described behaviour of the 35 signal is there a general recommendation to apply this correction to all versions of the AMS instrument (AMS, ACSM, ToF ACSM etc)?

---

## Author Comment (AC1) · 31 Jul 2020

**Improved chloride quantification in quadrupole aerosol chemical speciation monitors (Q-ACSMs)**

Anna K. Tobler[1], Alicja Skiba[2], Dongyu S. Wang[1], Philip Croteau[3], Katarzyna Styszko[4], Jarosław Nęcki[2], Urs Baltensperger[1], Jay G. Slowik[1] and André S. H. Prévôt[1]

[1]Laboratory of Atmospheric Chemistry, Paul Scherrer Institute, 5232 Villigen PSI, Switzerland
[2] AGH University of Science and Technology, Faculty of Physics and Applied Computer Science, Department of Applied Nuclear Physics, 30-059 Krakow, Poland
[3]Aerodyne Research, Inc, Billerica, Massachusetts, USA
[4]AGH University of Science and Technology, Faculty of Energy and Fuels, Department of Coal Chemistry and Environmental Sciences, 30-059 Krakow, Poland

*Correspondence to:* Jay G. Slowik (jay.slowik@psi.ch) and André S.H. Prévôt (andre.prevot@psi.ch)

**Author's response to anonymous Referee #1**

We thank anonymous Referee #1 for the careful revision and useful comments, which helped improve the quality of the manuscript. The referees' original comments (in *italic*) is followed by the author's answer (in regular typeset). Changes to the manuscript are indicated in green font.

*The manuscript provides the pathway for chloride quantification in the ACSM measurements. It is very timely and needed paper as, currently, despite the caveats listed in the paper, the Cl quantification is normally taken for granted. The manuscript is very well structured and nicely written, real pleasure to read.*

1.1) *However, I have a major concern with the lack of method validation. The corrected Cl data were not compared or validated to anything. Indeed, the signal does become positive after the corrections, however, there is no indication that that positive signal is quantitative. Since it is very important to show that the method works quantitatively, I would strongly suggest including a corrected Cl comparison with an independent measurement. Speciated Cl measurements would be ideal, but might not be readily available, so, at least, Cl an improvement of total volume/mass measured with ACSM and independent instrument/ instruments should be shown. Subject to this validation, I deem this manuscript suitable for the AMT.*

There are no online mass concentrations measured at AGH where the ACSM was installed. However, during a short period of time (15 March to 10 April 2019), an Xact 625i® Ambient Metals Monitor was measuring the elemental concentrations in ambient aerosols by X-ray fluorescence next to the ACSM in one hour resolution. The Xact was equipped with an automated alternating $PM_{2.5}$ and $PM_{10}$ inlet. For the following, we present a comparison of the ACSM measurements and selected $PM_{2.5}$ elemental concentrations.

As presented in Figure R1 a and b, the correlation of the chloride measurements by the ACSM and the Xact noticeably improves ($R^2$ increases from 0.35 to 0.94) when the proposed correction is applied. As a comparison, similar correlation is observed between the ACSM sulfate measurement and the Xact sulfur measurement. While the difference in absolute concentration likely represent a

general calibration issue from the ACSM or the Xact (i.e. absolute sensitivity), the improved correlation demonstrates that the correction is working very well. Discrepancies between the absolute concentrations between the ACSM and the Xact could be additionally caused by uncertainties in the collection efficiency ACSM (estimated CE = 0.5 according to Middlebrook et al. (2012)) or loss of semi-volatile chloride from Xact during the sampling collection.

[Figure]

**Figure R1.** Comparison of the chloride (original fragmentation table (a) and after correction (b)) and sulfate (c) concentrations measured by the ACSM with chloride and sulfur measurements of the Xact, measured between 15 March and 10 April 2019.

1.2) *Another important aspect, but, maybe, not as crucial as the one above, is higher than standard vaporiser temperature for this instrument. Why was this implemented, how does this compare to the standard t-re measurements? There are some indications, that 720C might still be comparable to standard 600C for some m/z, but better discussion around this is required. With some information on why this temperature was selected provided in the methods section as well. Were ambient measurements performed at this temperature as well?*

During the initial setup, the vaporizer voltage was set to 7.7 V in order to reach a vaporizer temperature of 600 °C as read by the thermocouple. In retrospect, the higher than usual voltage required reaching the target temperature likely stem from issues with thermocouple placement or contact with the vaporizer. To ensure data consistency throughout the campaign, the voltage was kept at 7.7 V for the ambient measurements as well as all the calibrations and chloride salt experiments (except when noted otherwise for the temperature dependencies).

The standard vaporizer temperature of 600 °C is chosen as a compromise between efficient vaporization of $(NH_4)_2SO_4$ and high ionization efficiency of $NH_4NO_3$ and organics. This is illustrated in Figure R2, which shows the measured ionization efficiency of $(NH_4)_2SO_4$, $NH_4NO_3$, and malonic acid as a function of vaporizer current and temperature. The boxed region denotes the temperature at which the above criteria are optimally balanced. Using a higher temperature for the vaporizer would likely lead to a decreased ionization efficiency for $NH_4NO_3$, but this will not bias the measurements because calibrations and ambient measurement were performed under identical instrument conditions. Therefore, the measurements should be comparable to measurements done with a vaporizer temperature of 600 °C.

[Figure]

**Figure R2.** IPP measurements for $NH_4NO_3$, $(NH_4)_2SO_4$ and malonic acid to determine the ideal vaporizer temperature (from: 11th AMS Users Meeting, Hyytiälä, Finland, Sept. 4-6, 2010, http://cires1.colorado.edu/jimenez-group/wiki/index.php/AMSUsrMtgs).

For more clarification we added in the manuscript:

line 85: While using a higher temperature for the vaporizer could lead to a decreased response for $NH_4NO_3$, this will also be reflected in the calibrations as these were done under the same conditions. Therefore, the measurements should be comparable to measurements done with a vaporizer temperature of 600 °C.

*1.3)    Finally, the assumption that it was indeed NH4Cl contributing to Cl signal is still not fully convincing. Better discussion on NH4Cl origins and potential sources in this region is required, also discussing the potential lack of other salts (why other salts are not likely) in the region. Correlation with Na is still significant, why?*

As described in the manuscript (line 40), $NH_4Cl$ can form from HCl or particulate chloride. A major source for such emissions are refuse incineration and coal combustion. The most common chloride salts besides $NH_4Cl$ are typically NaCl and KCl, which are characterized in the experiments described in the manuscript. Other chloride salts such as $MgCl_2$ and $CaCl_2$ can also be found in the particle phase but are undetectable by ACSM. Typical sources for NaCl can include sea salt. However, the distance to sea is > 480 km, so this source is unlikely to be significant. There is a salt mine southeast of Krakow, which could be a source for coarse-mode chloride. Biomass burning can be a source for KCl, depending on the type of biomass material; however, conversion of KCl to $KNO_3$ and $K_2SO_4$ in the atmosphere is also likely (Li et al., 2016).

We added to the manuscript:

line 219: Typical NaCl sources such as sea salt are unlikely (distance to sea > 480 km). The salt mine southeast of Krakow is a potential source of chloride, but only in the coarse mode. Biomass burning

can be a source for KCl, depending on the type of biomass material; however, conversion of KCl to $KNO_3$ and $K_2SO_4$ in the atmosphere is likely (Li et al., 2016).

In the manuscript, we present the correlation of *m/z* 36 with fragments from NaCl (*m/z* 23 and *m/z* 58) and KCl (*m/z* 39 and *m/z* 74). In addition, we here report the correlation of *m/z* 23 with the ACSM species. The correlation of chloride with $NH_4$ is significantly higher than its correlation with *m/z* 23. Although the correlation of *m/z* 23 could still be regarded as significant, the highest correlation of this ion is with $NH_4$. The correlation with the other species are all rather similar. The observed vaporization time scale of ambient chloride, which is consistent with that of $NH_4Cl$, suggests that the contribution of ambient NaCl is likely minor.

[Figure]

**Figure R3.** Correlation matrix of *m/z* 23 and the ACSM species.

*1.5) Minor:*

*Line 79: provide details for salts (sources, purity)*

The requested information was added to the manuscript:

line 79: Ammonium chloride ($NH_4Cl$, ≤ 100 %, Merck), sodium chloride (NaCl, ≥99.5 %, Fluka) and potassium chloride (KCl, ≥99.5 %, Merck) were separately dissolved in ultrapure water.

line 89: Calibrations were performed with the same setup using aqueous solutions of ammonium nitrate ($NH_4NO_3$, ≥99.5 %, Fluka), ammonium sulfate (($NH_4)_2SO_4$, ≥99.5 %, Fluka) and ammonium chloride ($NH_4Cl$), […]

*Line 82: provide info for drying (type of dryer, humidity after drying, RH stability)*

We clarified the information:

Mono-disperse particles with a diameter of 300 nm were generated using a Topas atomizer. Subsequently, the aerosol passed through a silica diffusion gel dryer, a krypton source as a bipolar charger and a custom-made differential mobility analyzer (DMA).

*Line 148: 'likely to suppress the m/z 36 signal and enhance the m/z 35 signal' – do you mean background, not diff? specify.*

We rephrased for clarity:

Line 148: Regardless of the reactions, any of these multi-step processes would be much more likely to suppress the *m/z* 36 signal and enhance the *m/z* 35 signal in the background. For example, it can be expected that tungsten oxide chlorides, produced by reaction with the vaporizer, will result in a $Cl^+$ signal rather than an $HCl^+$ signal.

*Lines 197 very high (40%) variability of RIEcl, was that temperature dependent? Discuss it. Also, discuss the exceptionally low RIEnh4. Was humidity stable during calibrations?*

Although the $RIE_{NH4}$ is typically between 3 and 6, this is strongly instrument-dependent as shown by Crenn et al. (2015) in the ACSM intercomparison study where they observed values between 3.17 and 14.72 (jump scan). In the second intercomparison campaign (Freney et al., 2019), values between 2.9 and 7.6 (full scan) were reported. Our value ($RIE_{NH4} = 2.43$) are close to the lower end of this and are relatively consistent and stable throughout the campaign. During the calibration the RH was not monitored. However, from independent experiments with the dryer, it can be assumed that the aerosol was well dried.

The RIE_Chl' is slightly less stable than $RIE_{NH4}$ and $RIE_{SO4}$, however, this likely reflects the lower signal intensity during calibration, as only the HCl signal is taken into account. Since the signal to noise ratio is inherently lower, it makes sense that the RIE_Chl' is a bit more uncertain. In addition, the value we report includes also a filament switch from filament 1 to filament 2.

**Author's response to anonymous Referee #2**

We thank anonymous Referee #2 for the careful revision and useful comments, which helped improve the quality of the manuscript. The referees' original comments (in *italic*) is followed by the author's answer (in regular typeset). Changes to the manuscript are indicated in green font.

*General Comments:*

*The manuscript by Tobler et al. focused on particulate chloride detection and quantification issues observed for some quadrupole aerosol chemical speciation monitors, which presented an approach to correcting the chloride concentration. This is an important and necessary work, which can be applied for measurements in environment where chloride is dominated by NH4Cl Overall the paper is well written. I recommend acceptance for publication on AMT after minor revisions.*

*Specific comments:*

2.1)   *What is the reason for setting the voltage to 7.7 V (line 83, 84)? Please elaborate it. A similar question was raised by Referee #1.*

The original setting of 7.7 V was due to miscommunication at the beginning of the campaign and was retained after the issue was discovered to maintain data consistency. It is unlikely to significantly affect the results presented here, as discussed in response to comment 1.2.

*2.2)*  *Please consider placing the high resolution peak fit of m/z 23, 39, 58 and 74 (line 208-215) in the supplementary.*

As described in the manuscript, interferences with organics are possible and highly likely for *m/z* 39, 58 and 74. However, all the measurements were done with a Q-ACSM and therefore only UMR data is available.

*2.3)*  *Before March 2018, the chloride shows the positive concentration (Fig.2), please compare this with the chloride after recalculation based on fragmentation table adjustments, and elaborate the error margin.*

Between 8 January and 15 February 2018, positive chloride concentrations were reported by the instrument using the standard fragmentation table. During this period, we estimate an average error of 26 % when using the standard fragmentation table.

[Figure]

**Figure R4.** Time series of the chloride concentrations during 8 January and 15 February 2018, based on the original and corrected fragmentation table.

The comparison of the chloride concentrations based on the original and adapted fragmentation table has been added to the supplement, with the following text added to the manuscript:

line 223: An average error of 26 % is estimated using the standard fragmentation table instead of the here proposed correction and calibration for the time between 8 January and 15 February 2018, when positive chloride concentrations were reported with the standard fragmentation table and RIE$_{Chl}$ (Fig. S3).

*2.4)*  *I suggest that the sample/filter cycle in Fig.3 and 4 be shaded as an indicator, just like Fig.2.*

We agree and we have also updated the corresponding supplementary figures as suggested. The updated plots and legends in the main text are shown below.

[Figure]

**Figure 3. Time series of the signal of (a)** *m/z* **35 and (b)** *m/z* **36 with 1 s resolution over a simulated sample (dark grey)/filter (light grey) cycle for NH4Cl (pink), NaCl (green) and KCl (orange). The maximum and minimum signals of NaCl and KCl are normalized to the maximum and minimum of NH4Cl.**

[Figure]

**Figure 4. Highly time-resolved signal of (a)** *m/z 35* **and (b)** *m/z 36* **as a function of time at different vaporizer voltages (i.e. temperatures) over a simulated sample (dark grey)/filter (light grey) cycle for NH4Cl.**

**Author's response to anonymous Referee #3**

We thank anonymous Referee #3 for the careful revision and useful comments, which helped improve the quality of the manuscript. The referees' original comments (in *italic*) is followed by the author's answer (in regular typeset). Changes to the manuscript are indicated in green font.

*This manuscript describes the presence of a negative chloride signal measured in the widely used aerosol chemical speciation monitor (ACSM) instrument. This is an issue that has been observed in several instruments and up to now has not been formally addressed. In this work, the authors present long term observations of this artefact and perform additional detailed tests on instrument performance. The authors illustrated than this negative signal is essentially an artefact (stating that no other information regarding*

*the source of Cl can be extracted), and propose a simple correction to the standard fragmentation table to account for it. Given the widespread use of the ACSM, this type of work is essential to providing homogenous measurements among all operating instruments. This manuscript is well written with clear and concise text and well-presented figures. Although I recommend the manuscript for publication, I have some questions and comments below.*

3.1) *It is stated in the manuscript that there are a number of uncertainties related to how this artefact manifests itself in different instruments. The authors cite a personal communication whereby changing out the filament removes this artefact. Can the authors provide more information on this; does the artefact return after some time or is this artefact only present in instruments with an iridium filament?*

The Q-ACSM in Romania reported apparent negative chloride concentrations from September 2017 until December 2018, when the iridium filament was replaced with a tungsten filament. Afterwards, apparent negative chloride concentrations were not measured.

We have evidence to believe that this artefact can also be present with tungsten filaments. However, it is more likely with iridium filaments to report apparent negative concentrations while with the tungsten filament we are not aware of reported apparent negative concentrations. As discussed in the manuscript, the *m/z* 35 signal is impacted by the slow-vaporization behavior also with a tungsten filament and therefore not accurately represented.

3.2) *Was this instrument newly installed at the start of sampling. Could the inversion of Cl be a result of the build-up of material (the total PM concentrations observed during the field campaign are very high)? The appearance of the negative m/z-35 was very sudden, did it correspond to any changes in meteorological conditions?*

The instrument was newly installed at the measurement side in Krakow. However, the instrument has been used in other campaigns before, including the ACSM intercomparisons in 2013 and 2016 as well as in a campaign in Cabauw (NE).

As shown in Fig. 1d in the manuscript, the instrument response changes over time. The instrument history and the current state of the vaporizer clearly influence the magnitude of this outcome. The changed behavior is likely influenced by changes in the surface chemistry of the vaporizer due to Cl exposure (Drewnick et al., 2015). However, the exact mechanism of this not fully clear.

As described in the manuscript, the appearance of apparent negative chloride is due to the slow-vaporizing nature of *m/z* 35. We could not find indications that a change in meteorological conditions is related to the appearance of apparent negative chloride concentrations (Figure R5). We added the full meteorological data to the supplementary and added to the manuscript:

line 103: The change to apparent negative concentrations cannot be related to a change in meteorological conditions (Fig. S1).

[Figure]

**Figure R5.** Meteorological data (relative humidity, rainfall, wind speed, wind direction and temperature) does not imply that the apparent negative chloride is related to a change in meteorological parameters.

3.3) *A slow decay and slow build-up (as well as an artefact that disappears when the filament was changed) would suggest that material is built up on the vaporizer and the surrounding area. Was the filament changed (in this instrument) after the experiment to investigate this? Do you have an approximate temperature range for your experiments (that correspond to the voltages used)? Were any improvements observed after heating the vaporizer to > 800 C over extended periods of time? Line 110 (Figure 2): Is it possible to change the instrument settings so the sampling periods correspond to the end of the filter and sample run. This would better represent species that slowly build up and slowly decay?*

The ACSM is typically equipped with two filaments of the same type. During the campaign, we switched from filament 1 to filament 2 on 3 January 2019 following the failure of filament 1 . A physical exchange of intact filaments did not take place during or after the campaign. Based on the vaporizer temperature calibration performed by the manufacturer (Aerodyne Research, Inc.) prior to the delivery of the instrument, the experiments presented in the manuscript span a temperature range from 600 °C to 770 °C. When heating the vaporizer to > 800 °C overnight, no significant improvements were observed.

The issue of slow-vaporizing species is not unique to chloride or particular ions; it potentially affects all ions of the measured mass range and is the reason why all instrument calibrations are now performed in full scan mode (i.e., using the same timing scheme as the standard measurement).

The ACSM does allow changing the scan settings to enable for example longer filter/sample periods would be longer, down instead of up scanning, or that a delay after switching between the two modes. However, changing the instrument settings enough to get rid of this issue, this would result in a considerably long waiting time so that the instrument time resolution and signal to noise ratio would be considerably compromised. Therefore, this is not a desirable approach for general use.

*3.4)* *Why is the Chloride (m/z 35) signal in the negative so much larger than in the positive? As is observed in the latter part of 2018 and early 2019. At the very end of the sample period, it appears that the total reported Cl returned positive again, is this the case?*

In general, the signal intensity at $m/z$ 35 is higher compared to $m/z$ 36 ($Cl^+$ formation is favored over $HCl^+$), so any negative/underestimation artifacts will also be larger. The reported chloride concentration based on the standard fragmentation table is slightly positive again towards the end of the campaign, as a result of a more dominant $m/z$ 36 compared to $m/z$ 35. However, the signal at $m/z$ 35 never recovers to positive values after the reported negative values in February 2018 and leads to underestimation of total chloride mass when the original fragmentation table is applied.

[Figure]

**Figure R6.** Signal of *m/z* 35 and *m/z* 36, together with the reported chloride concentrations (original fragmentation table) towards the end of the campaign. The signal at *m/z* 35 is still negative and can lead to significant underestimation of the total chloride mass when included in its calculation.

*3.5)* *During the 14 month sampling period what other instruments were sampling along- side the ACSM, e.g. number and size distribution, filter measurements etc. Were any complementary measurements of refractory species made during this time.*

There are only very limited external measurements available throughout the campaign. There are additional measurements of eBC measured by the aethalometer AE33. Although this instrument should automatically compensate for loading effects, there are some issues with it during high pollution episodes. Additionally, elemental concentrations measured by an Xact 625i® Ambient Metals Monitor are available for a limited time (15 March to 10 April 2019).

3.6) *How did the measured ACSM total mass compare with the total mass measured by the SMPS (if present) during these sample periods (excluding the negative chloride peaks periods)? Are there indicators of the presence of refractory species during this time.*

The ACSM measurements were accompanied by eBC measurements using an aethalometer AE33 and from 15 March to 10 April 2019, an Xact 625i® Ambient Metals Monitor was measuring the elemental concentrations in ambient aerosols by X-ray fluorescence next to the ACSM. A comparison of two ACSM species and Xact metals is shown in response to comment 1.1).

3.7) *Line 213: were any correlations observed between m/z 35 and Na+ (m/z 23) and or K+ (39)?, could these peaks also have interference with species other than NaCl+ and KCl+? When the correction is applied, is all the NH4 measured accounted for by that predicted from Cl-, NO3-, and SO42- (in the form of NH4Cl, NH4NO3, and (NH4)2SO4 respectively).*

Little correlation between $m/z$ 35 and $Na^+$ ($m/z$ 23) and or $K^+$ ($m/z$ 39) can be observed ($R^2 = 0.04$ and 0.16, respectively). Typical HR-AMS measurements show no interferences at $m/z$ 23, whereas interferences by $C_3H_3^+$ at $m/z$ 39 can be expected (line 209 in the manuscript), with the possible addition of $C_2HN^+$, though these interferences cannot be characterized with UMR ACSM measurements.

Overall, the aerosol is mostly neutralized. The following plot shows the measured $NH_4$ versus the predicted $NH_4$, assuming $NH_4$ is fully neutralized by Chl, $NO_3$ and $SO_4$. For the plot, hourly averaged data was used and color-coded by the signal of $m/z$ 35.

[Figure]

**Figure R7.** Ion balance for the ACSM measurements (1 hour resolution). The measured $NH_4$ concentrations are on the y-axis, the predicted $NH_4$ concentrations ($NH_{4, pred} = 18 \times (NO_3/62 + 2 \times (SO_4 /96) + Chl /35.45)$ are on the x-axis. The grey line represents the 1:1 and corresponds to the neutralized aerosol. The points are color-coded based on the signal measured at $m/z$ 35.

*3.8)* *What recommendations should be given to data that is already submitted to data sets (e.g EBAS)?*

As shown in the manuscript, this correction is clearly important for environment with high chloride concentrations that are dominated by $NH_4Cl$ and improves the quantification significantly.

The proposed fragmentation table correction could be applied to any pre-existing dataset to assess qualitatively the trend of chloride. If chloride was included during calibrations, a quantitative estimation of $RIE_{Chl}$ and quantification of chloride mass is possible, assuming stable conditions in the instrument. For a majority of environments, the chloride contribution are likely minor (though this could be in part due to the negative chloride artefact). We expect the proposed correction to have only minor effects on the bulk non-refractory aerosol mass and composition derived from pre-existing data in most datasets, but strongly encourage the inclusion of chloride in future ACSM/AMS calibrations, as well as re-evaluation of chloride signal in existing datasets on an individual basis.

*3.9)* *Given the described behaviour of the 35 signal is there a general recommendation to apply this correction to all versions of the AMS instrument (AMS, ACSM, ToF ACSM etc)?*

It can be assumed that a similar behavior can also be found for other types of AMS instruments (line 216 onwards). In general, it is recommended to use fast vaporizing species, consistent with Ovadnevaite et al. (2012), where the $NaCl^+$ ion was suggested as a surrogate for sea salt.

While this presented technique for the chloride quantification could be applied to other versions of the AMS, more characterization of those systems would be needed to for a general assessment. The effect is reduced in the AMS and ToF-ACSM systems because they do not rely on the very slow scanning we use in the Q-ACSM, so their *m/z 35* and *m/z 36* measurements represent averages over the entire open or closed, sample or filter time rather than a single point in time along the rise/decay curve.

In addition, there are circumstances under which the effect is small enough and instrument performance is good enough, so that the answers are equivalent and there can be a trade-off between using the fast vaporizing species and the signal to noise ratio in the instrument. For instrument that do a chloride calibration, it is possible that they are compensating the behavior of *m/z* 35 by use of a different RIE, assuming that the instrument conditions are stable throughout the calibration and campaign period.

Crenn, V., Sciare, J., Croteau, P. L., Verlhac, S., Frohlich, R., Belis, C. A., Aas, W., Auml;ijala, M., Alastuey, A., Artinano, B., Baisnee, D., Bonnaire, N., Bressi, M., Canagaratna, M., Canonaco, F., Carbone, C., Cavalli, F., Coz, E., Cubison, M. J., Esser-Gietl, J. K., Green, D. C., Gros, V., Heikkinen, L., Herrmann, H., Lunder, C., Minguillon, M. C., Mocnik, G., O'Dowd, C. D., Ovadnevaite, J., Petit, J. E., Petralia, E., Poulain, L., Priestman, M., Riffault, V., Ripoll, A., Sarda-Esteve, R., Slowik, J. G., Setyan, A., Wiedensohler, A., Baltensperger, U., Prevot, A. S. H., Jayne, J. T., and Favez, O.: ACTRIS ACSM intercomparison - Part 1: Reproducibility of concentration and fragment results from 13 individual Quadrupole Aerosol Chemical Speciation Monitors (Q-ACSM) and consistency with co-located instruments, Atmos. Meas. Tech., 8, 5063-5087, https://doi.org/10.5194/amt-8-5063-2015, 2015.

Drewnick, F., Diesch, J. M., Faber, P., and Borrmann, S.: Aerosol mass spectrometry: particle-vaporizer interactions and their consequences for the measurements, Atmos. Meas. Tech., 8, 3811-3830, https://doi.org/10.5194/amt-8-3811-2015, 2015.

Freney, E., Zhang, Y. J., Croteau, P., Amodeo, T., Williams, L., Truong, F., Petit, J. E., Sciare, J., Sarda-Esteve, R., Bonnaire, N., Arumae, T., Aurela, M., Bougiatioti, A., Mihalopoulos, N., Coz, E., Artinano, B., Crenn, V., Elste, T., Heikkinen, L., Poulain, L., Wiedensohler, A., Herrmann, H., Priestman, M., Alastuey, A., Stavroulas, I., Tobler, A., Vasilescu, J., Zanca, N., Canagaratna, M., Carbone, C., Flentje, H., Green, D., Maasikmets, M., Marmureanu, L., Minguillon, M. C., Prevot, A. S. H., Gros, V., Jayne, J., and Favez, O.: The second ACTRIS inter-comparison (2016) for Aerosol Chemical Speciation Monitors (ACSM): Calibration protocols and instrument performance evaluations, Aerosol Sci. Technol., https://doi.org/10.1080/02786826.2019.1608901, 2019.

Li, W., Shao, L., Zhang, D., Ro, C.-U., Hu, M., Bi, X., Geng, H., Matsuki, A., Niu, H., and Chen, J.: A review of single aerosol particle studies in the atmosphere of East Asia: morphology, mixing state, source, and heterogeneous reactions, Journal of Cleaner Production, 112, 1330-1349, https://doi.org/10.1016/j.jclepro.2015.04.050, 2016.

Middlebrook, A. M., Bahreini, R., Jimenez, J. L., and Canagaratna, M. R.: Evaluation of Composition-Dependent Collection Efficiencies for the Aerodyne Aerosol Mass Spectrometer using Field Data, Aerosol Sci. Technol., 46, 258-271, https://doi.org/10.1080/02786826.2011.620041, 2012.

Ovadnevaite, J., Ceburnis, D., Canagaratna, M., Berresheim, H., Bialek, J., Martucci, G., Worsnop, D. R., and O'Dowd, C.: On the effect of wind speed on submicron sea salt mass concentrations and source fluxes, J. Geophys. Res.-Atmos., 117, https://doi.org/10.1029/2011jd017379, 2012.